# Colorimetric Indicator Based on Gold Nanoparticles and Sodium Alginate for Monitoring Fish Spoilage

**DOI:** 10.3390/polym16060829

**Published:** 2024-03-17

**Authors:** Lissage Pierre, Julio Elías Bruna Bugueño, Patricio Alejandro Leyton Bongiorno, Alejandra Torres Mediano, Francisco Javier Rodríguez-Mercado

**Affiliations:** 1Packaging Innovation Center (LABEN), Food Science and Technology Department (DECYTAL), Technological Faculty, University of Santiago de Chile (USACH), Ave. Víctor Jara 3769, Santiago 9170124, Chile; alejandra.torresm@usach.cl (A.T.M.); francisco.rodriguez.m@usach.cl (F.J.R.-M.); 2Center for the Development of Nanoscience and Nanotechnology (CEDENNA), University of Santiago de Chile (USACH), Santiago 9170022, Chile; 3Instituto de Química, Pontificia Universidad Católica de Valparaíso, Valparaíso 2340025, Chile; patricio.leyton@pucv.cl

**Keywords:** gold nanoparticles, polymer, colorimetric indicator, sodium alginate, biogenic amines, ethylenediamine

## Abstract

In this work, a colorimetric indicator based on gold nanoparticles (AuNP) and a biodegradable and eco-friendly polymer (sodium alginate, Alg.), was developed for the real-time detection of fish spoilage products. The AuNPs and the colorimetric indicator were characterized using UV-VIS, FTIR spectroscopies, TGA, DSC, XRD, TEM, and colorimetry. The UV-VIS spectrum and TEM showed the successful synthesis, the spherical shape, and the size of AuNPs. The results indicated color changes of the indicator in packaged fish on day 9 of storage at a refrigerated temperature (5 °C. These results showed the successful application of the colorimetric indicator in the detection of TVB-N in packaged fish.

## 1. Introduction

Fish is an excellent source of nutrients and is easily digestible in the human diet [1] because it contains healthy fats, omega-3 (EPA, DHA), high-quality protein, iron, phosphorus, zinc, selenium, vitamins, and minerals. However, it is highly perishable, vulnerable, and very prone to deterioration due to enzymatic reactions or microbial contamination during handling, distribution, and/or storage, leading to the formation of different metabolic products, such as alcohols, ketones, some aldehydes, organic acids, and sulfides, which usually occurs after the death of the fish. Storage in inappropriate places and/or poor hygiene conditions will lead to deterioration and the formation of biogenic amines (BA) [2], among which the most abundant are putrescine, cadaverine, tyramine, and histamine, some of which are dangerous to the human organism [3]. In the same way, the other factors that play an important role in the deterioration of fish through the consequent production of BA are bacteria, mainly Gram-positive and Gram-negative bacteria. In general, they are located in different parts of the fish body, particularly in the skin, gills, or gastrointestinal tract [4,5,6], and can also spread to the muscle mass during evisceration through the rupture or loss of gastric contents. The species most frequently found in this decomposition process are *Enterobacteriaceae*, including mesophilic and psychrotolerant bacteria, such as *Morganella*, *Enterobacter*, and Gram-negative bacteria of the *Hafniaceae* family: *Hafnia*, *Proteus*, and *Photobacteria* [7]. Furthermore, the genus *Pseudomonas* and lactic acid bacteria belonging to the genera *Lactobacillus* and *Enterococcus* can cause BA formation [8].

In this sense, in recent times, various techniques have been used to detect the quality of fish, but they are mainly based on the analysis of its structure (tenderness, color, texture, etc.), using methods of counting bacteria, determining the total volatile basic nitrogen (TVB-N), and measuring pH values. Other techniques, such as optical spectroscopy, Nuclear Magnetic Resonance (NMR), Fourier-transform infrared spectroscopy (FTIR), and gas chromatography–mass spectrometry (GC/MS), are also used for evaluation. However, there are disadvantages associated with the use of these techniques because, in some cases, the sample is destroyed; a complex, slow, expensive sample preparation is required; and, in general, an expert is required to execute the methods [9]. Therefore, the intelligent packaging system is elaborated by the incorporation of a device that can interact internally or externally, monitoring the changes that may occur in the packaged product. This can be separated or listed into three groups: (1) indicators are established to provide information about the quality of the products to consumers; (2) sensors are prepared to detect specific analytes in packaged foods; and (3) data carriers (including barcodes and radio frequency identification labels) are designed to carry out traceability and/or monitoring in the food supply chain [10].

In this context, Alg. is an anionic polymer produced by brown algae and bacteria. It is a biocompatible, biodegradable, non-toxic, low-cost, and readily available polymer. It consists of α-L-guluronic acid (G) and β-D-mannuronic acid (M) residues linked linearly by 1,4-glycosidic bonds [11]. It is widely used in various fields and in the food industry because it is capable of producing strong gels in the presence of metal cations. Thus, it is considered convenient to develop an indicator based on the metallic nanoparticles inserted in Alg. In this sense, colorimetric indicators are valuable and interesting for application due to their low cost, their ease of use, their simplicity, and, above all, they offer high legibility with the naked eye [12]. Therefore, in previous studies, researchers have developed a colorimetric label based on bacterial cellulose with the incorporation of grape anthocyanin that allows the monitoring of the freshness of stored minced meat [13]. Wang et al. [14] reported the development of an elementary colorimetric sensor that can be used as a cheaper indicator to detect the freshness of fish using PANI films, which can be regenerated using acid solution. Furthermore, a pH-sensitive sensor based on cellulose-modified polyvinyl alcohol (PVA) was developed. Its evaluation in deteriorated fish produced color changes [15].

On the other hand, other groups of researchers have developed an indicator matrix with 16 diverse detection components to monitor fish spoilage [16]. Although the colorimetric indicator based on pH-sensitive dyes is a simple way to control the quality of food, which can be observed with the naked eye, it is essential to develop a new detection system based on metal nanoparticles inserted in a biodegradable and eco-friendly polymer for the real-time monitoring of fish deterioration. In this research work, a colorimetric indicator based on gold nanoparticles and a natural polymer was developed, capable of indicating, through a color change, when the fish was unsuitable for consumption.

## 2. Materials and Methods

### 2.1. Materials

To obtain gold nanoparticles (AuNP), sodium tetrachloroaurate (III) dihydrate (NaAuCl_4_·2H_2_O) 99% CAS No. 13874-02-07 and trisodium citrate dihydrate (NaCit) (C_6_H_5_Na_3_O_7_·2H_2_O) CAS No. 6132-04-3 were used as reagents; they were purchased from Sigma-Aldrich and used without further purification and distilled water.

### 2.2. Synthesis of Gold Nanoparticles (AuNP)

Gold nanoparticles (AuNPs) were synthesized using the method described by El-Nour et al. [17] with some modifications. Briefly, 40 mL of distilled water was poured into a 100 mL flask and heated until boiling, then 1 mL of 1% trisodium citrate was gradually added with continuous stirring. Then, 100 µL of 1% sodium tetrachloroaurate (III) dihydrate solution was added and the solution was stirred and heated until the color of the solution changed from yellow to deep red at approximately 5 min, indicating the formation of AuNP. Gold nanoparticles were gradually formed as the citrate reduced Au_3+_ to Au_0_ as indicated by the red color. The AuNP solution was cooled down at room temperature and stored at 4 °C until further use.

### 2.3. Colorimetric Indicator Preparation

Colorimetric indicators were prepared according to the method described by Dudnik et al. [18]. Briefly, an aqueous solution was added to Alg. (2 wt%) and stirred at 1000 rpm for 3 h or until complete dissolution. Then, 3 mL of dispersion of gold nanoparticles was added to 1 mL of Alg. with continuous stirring. Then, the colloidal suspension was cast into a Petri dish containing 0.10 mol/L solution of calcium chloride to obtain an Alg. gel and allowed to dry in ambient conditions. The prepared colorimetric indicators were then packed in a 20 mL screw-cap vial and stored at room temperature until further use.

### 2.4. Characterization

The synthesized AuNP was characterized using the UV-VIS spectrophotometer Spectroquant^®^ pharo 300M, from Merck KGaA, Darmstadt, Germany in the range of 300–800 nm. The samples were placed in the cuvette, and the UV-VIS spectrum was measured at different time intervals.

An FTIR analysis was performed to characterize the chemical structure of the materials (Alg. and AuNP). For this, a colloidal solution of nanoparticles was concentrated under a vacuum (vacuum system) in a rotary evaporator (Heidolph, Schwabach, Germany) and dried using lyophilization. For the analysis, pellets were obtained by mixing the dry gold nanoparticles with KBr 1.0 wt% of the sample. The FTIR equipment model, Alpha II Bruker, Waltham, MA, USA was used in the following conditions: a range of 4000–500 1/cm and a resolution of 4 1/cm, which was obtained after cumulating 64 scans to determine the FTIR of the AuNP and the indicators, respectively. To determine the thermal properties of gold nanoparticles (AuNP) and the colorimetric indicator, a thermogravimetric analysis was performed using a thermogravimetric analyzer (METTLER TOLEDO Gas Controller GC20 Stare system TGA/DSC) at a rate of 10 °C min^−1^ under nitrogen gas, flowing at 20 mL min^−1^ in the range of 30–800 °C. The DSC analyses were carried out using (DSC1 equipment, model STAR System 822, MA, USA) operating at the following conditions: a heating rate of 10 °C min^−1^, a nitrogen flow rate of 20 mL min^−1^, and a temperature range from 0 to 400 °C and 0–200 °C for the AuNP and indicators, respectively.

An X-ray diffraction (XRD) analysis was carried out using an X-ray diffractometer (D8 Advance, Bruker, Germany) in the range of 2–80° at an angle of 2θ with CuKα radiation. The tube current and voltage were 30 mA and 40 kV, respectively, and the scan rate was 1°/min. The morphology of the AuNP was studied using Transmission Electron Microscopy (TEM) images (Talos F200X (Thermo Fisher Scientific)), Waltham, MA, USA. The microscope was operated at an accelerating voltage of 200 kV. The sample (10 μL) was mounted on a copper grid covered by a carbon–formvar film and allowed to dry at room temperature for 24 h before the TEM analysis.

### 2.5. Detection Capacity and Colorimetric Analysis

The detection capacity of gold nanoparticles was evaluated by preparing a stock solution of ethylenediamine (ETD) with a concentration of 2000 ppm (mg/L), from which the following concentrations were obtained: 10–400 ppm. A colorimetric analysis of the AuNP and the colorimetric indicator was performed by obtaining photographs with a digital camera. The photographs obtained were evaluated using the ImageJ software (2.9.0/1.53t/Java 1. 8. 0_322 (64-bit)), determining the color parameters (*L**, *a**, and *b**) based on the CIELab color system, and the total color difference according to the following formula: $$∆E=\sqrt{{\left(∆L*\right)}^{2}+{\left(∆a*\right)}^{2}+{\left(∆b*\right)}^{2}}$$

*L** = lightness (0 = black, 100 = white), +*b** (yellow), −*b* (blue), +*a** (red), −*a** (green).

### 2.6. Evaluation of Indicators under Simulated Conditions (In Vitro)

The in vitro evaluation of the colorimetric indicator was performed using ETD, according the method described by Zhai et al. [19] with slight modifications. For this, the aforementioned indicator was taken and placed in a clean and smooth plastic Petri dish with a 90 mm diameter and with the ETD solution at room temperature, observing the possible color changes of the indicator. Then, the indicators were peeled from the Petri dishes and observed in an optical microscope, taking photos with a Samsung smartphone (Galaxy A21S). Then, the total color differences were analyzed using ImageJ software in triplicate and the data were processed using the CIELAB system to determine the color parameters (*L**, *a**, and *b**) [20].

### 2.7. Evaluation of the Indicator in Real Packaging Conditions

For the evaluation of the indicator in real packaging conditions, Atlantic salmon was used as a model food, which was purchased in the local trade. For this, the fish were aseptically filleted with a knife to obtain more-or-less uniform rectangular samples of equal weight (approximately 50 g). All the fish samples were introduced into sterile vacuum stomacher bags, in which the colorimetric indicator to be evaluated was inserted, using a polyethylene/polyamine support so that the indicator was kept in direct contact with the muscle food. The stomacher bags were stored at a temperature of 5 °C for 9 days to monitor the spoilage and the color change of the indicator. At specified intervals during the storage period, the samples were analyzed for any chemical or microbial changes. Reference stomacher bags containing fish samples without indicators were also maintained for the same experimental period as those containing indicators.

### 2.8. Characterization of Fish Samples during Storage

In all the fish samples, the pH and the total volatile basic nitrogen (TVB-N) were determined in duplicate (*n* = 2). The pH measurements were collected using a Meat pH Tester (Hanna Instruments, Smithfield, RI, USA). The calibration was previously carried out using buffers 4 and 7, then the pH was measured by inserting the tip, in the form of a glass cone, inside the fish and obtaining the corresponding values.

### 2.9. Determination of Total Volatile Basic Nitrogen (TVB-N)

The total volatile basic nitrogen (TVB-N) values of salmon were determined in accordance with the Chilean standard norm NCh 2668:2018 for hydrobiological products. Briefly, 10 g of the fish sample, 2 g of magnesium oxide, and 100 mL of water were added to the distillation flask, followed by the distillation of the mixture. The distillate was taken up in 25 mL of a 3% *m*/*v* aqueous solution of boric acid and 5–7 drops of Tashiro’s indicator (mixed indicator composed of a solution of methylene blue (0.1%) and methyl red (0.03%) in ethanol or methanol). Then, the boric acid solution was titrated to the endpoint with 0.1 N sulfuric acid. The TVB-N content (mg of N/100 g) of the fish sample was determined as follows:$$TVB-N=\frac{mL\;H_{2}{SO}_{4}\;0.1N\times 1.4\times 100}{Sample\;weight\left(g\right)}$$

### 2.10. Microbiological Analysis

As the fish spoilage was being monitored using colorimetric indicators, samples were subjected to a microbiological analysis at regular intervals. For the total microbial count determination, 25 g of the salmon sample was added to 225 mL of 0.1% *m*/*v* peptone water. The mixture was homogenized in a homogenizer for 1 min and the 0.1 dilution was obtained. From this dilution, further decimal dilutions were prepared and plated on Petri dishes in the appropriate media. The enumeration of the total viable aerobic bacteria counts was performed according to the pour plate method, using plate count agar (PCA) purchased from Merck. The inoculated Petri dishes were incubated at 35 ± (1 °C) for 48 h to determine the mesophilic counts. The colony-forming unit (CFU) counts were expressed as CFU/g. All the microbiological experiments were performed in duplicate.

### 2.11. Statistical Analysis

The analyses and experiments were performed in triplicate and the data were evaluated using analysis of variance (ANOVA) (comparison of several samples) using the PROGRAM-STATGRAPHICS Centurion XIX.v.64, followed by Fisher’s Least Significant Difference (LSD) procedure. A probability level of *p* < 0.05 was considered statistically significant.

## 3. Results

### 3.1. Characterization of Gold Nanoparticles (AuNPs)

Gold nanoparticles were characterized using UV-VIS, transmission electron microscopy (TEM), colorimetric analysis, infrared spectroscopy (FTIR), thermogravimetric analysis (TGA), differential scanning calorimetry (DSC), and X-ray diffraction (XRD).

In Figure 1, a color change of the solution was observed, it changed from yellow to a red dispersion with the addition of trisodium citrate (NaCit), indicating the formation of gold nanoparticles [21]. In addition, it was possible to observe through the UV-VIS spectrum an absorption band at a wavelength of 520 nm in the red color of the spectrum that gives its hue; this phenomenon is related to the surface plasmon resonance (SPR) because the AuNP conduction electrons interact with incident photons to produce a resonance effect, manifested as SPR. This interaction depends on the size, the shape, and the composition of the metallic nanoparticles, as well as the type and content of the dispersion medium [22]. Similar results were obtained by [23].

Figure 2 shows the results of the detection capacity of the AuNPs by UV-VIS spectrum, observing two absorption peaks. The first peak was between 520 nm and 531 nm, which may be associated with the aggregation of the AuNPs caused by their contact with the ETD solution. The second absorption peak was located at a wavelength between 660 nm and 702 nm, which is associated with a change in the morphology of the AuNPs because its union with the ETD produces a strong interaction between particles, varying their shape and size, as well as their color [24]. Similar results were reported by Sun et al. [25]. It should be noted that the bathochromic shift of AuNP results in a large displacement towards the red region of the Localized Surface Plasmon Resonance (LSPR) peak [26] Furthermore, Mahatnirunkul et al. [27]. affirmed that when the analyte of interest binds to the surface of AuNPs, the Localized Surface Plasmon Resonance (LSPR) spectrum will shift to a longer wavelength. Similar results were reported by [28]. Additionally, the aggregation of gold nanoparticles, after their contact with ethylenediamine, occurred through the interaction of attractive van der Waals (VA) and repulsive Coulomb (VR) forces. Certainly, the gold nanoparticles in suspension remain stable when VR > VA. On the other hand, when VR < VA, the nanoparticles clump together [29]. It has been detected that a wide variety of factors intervene in this process, such as particle size, surface tension, and the electrical double layer, which have a high participation in reducing the stability of the NPs and their possible aggregation. However, in this case, the citrate ions adsorb on the surface of the prepared AuNPs, creating a negative surface charge that stabilizes the particles; the energy barrier was powerful enough to prevent a strong interaction between the particles and, with that, it prevented them from aggregating. However, the addition of ETD to NaCit-stabilized AuNPs disturbed the stability of the nanoparticles, leading to their aggregation. Additionally, we used ETD as a test molecule for the in vitro evaluation of the detection ability of AuNPs, because it is an analogous molecule to TVB-N, such as NH_3_, dimethylamine (DMA), and trimethylamine (TMA); they have in common the amino group, which contains nitrogen. Since biogenic amines are one of the main spoilage products of fish that we need to detect with the colorimetric indicator. 

Furthermore, when a higher AuNP extinction is observed, this is an indication of a higher concentration of nanoparticles. It should be noted that, as the nanoparticles clump together, they increase in size, gradually settling to the bottom and resulting in less extinction. Similar results were obtained in a previous study by Ranjan et al. [30].

In order to study the size and morphology of the synthesized AuNPs, TEM measurements were carried out. In Figure 3a, it can be observed that all the particles presented a homogeneous spherical shape, as expected for this type of nanoparticle [31]. Furthermore, it presented a wide size distribution from 10 to 16 and 50 nm, approximately. On the other hand, in Figure 3b, it is shown that by adding ETD, the AuNPs aggregated and formed clusters due to a strong interparticle interaction. In addition, in the UV-VIS spectrum, it was possible to corroborate a displacement of the wavelength towards the red region of the spectrum, indicating larger nanoparticles as demonstrated in the TEM analysis.

The FTIR analysis of the citrate-capped AuNPs is presented in Figure 4, where it is possible to observe the presence of the absorption bands at 1696, 1580, 1387, 1281, and 1079 1/cm. The band at 1696 1/cm is related to the COO^−^ group, originating due to the adsorption of the citrate anions on gold nanoparticles through the central carboxylate group [29]. The two bands with their peaks at around 1387 and 1580 1/cm, respectively, corresponded to the symmetric (*ν_s_*(COO^−^)) and antisymmetric (*ν*_as_(COO^−^)) stretching bands of the carboxyl groups of NaCit. For the citrate-capped AuNP, the *ν*_as_(COO^−^) peak was found to be largely a high-frequency shift from the original peak position. In addition, absorption bands were observed between 1281 and 1079 1/cm, which corresponded to the stretching vibration of the C–O bond, as well as the C–C bond of NaCit [32]. Similar results were reported elsewhere by [33]. The 3344 1/cm and 3446 1/cm absorption bands that appeared between 3600 and 3000 1/cm were related to the stretching vibration of the O–H bond of NaCit. On the other hand, changes in the stretching vibration of the O–H group of the citrate were observed with the addition of AuNP. This was due to the interaction of the nanoparticles surrounded by the NaCit, giving rise to the formation of a new absorption band at wavenumber 3344 1/cm. This indicated the disappearance of the citrate band (3446 cm^−1^), evidencing the presence of gold nanoparticles [34].

In this experiment, the AuNPs covered by a layer of negatively charged citrate ions were used to keep the nanoparticles well dispersed and stable in the colloidal solution. The addition of 10 ppm ETD in 1 mL of AuNP caused an aggregation of AuNPs (Figure 5 and Table 1), mainly affecting the extinction spectrum with color changes of the AuNPs with a total color difference of ΔΕ = 28.81. This aggregation could be due to an electrostatic interaction between the positively charged amine groups and the negatively charged trisodium citrate ion groups surrounding the AuNPs, or could also be the result of some exchange between the trisodium citrate ions and amines that probably can directly adhere to the AuNPs. These results indicated the ability of the metallic nanoparticles to detect fish spoilage products by properly interacting with the test molecule, causing the formation of larger nanoparticles with a subsequent color change in the dispersion; however, by adding 200 ppm of ETD in 1mL of AuNP, a ΔΕ = 36.9 was obtained. This indicates that the higher the concentration of ethylenediamine, the greater the color changes observed.

Figure 6 shows the TGA/DTGA of citrate-capped AuNP, where five mass losses were observed. The first mass loss corresponds to the release of water from the sample and began at (166 °C) to (250 °C), representing 7.83% of the initial mass. The second mass loss was between (256 °C) and (336.5 °C), with a peak at (288.5 °C) corresponding to a loss of 5.5%. The third endothermic peak was observed in the range of (375.3 °C) to (482.6 °C), with a peak at (437.3 °C) corresponding to a mass loss of 5.1%. The second and third peaks after the dehydration peak probably correspond to the degradation of the sample and/or the organic matter covering the trisodium citrate. The fourth and fifth mass losses were observed in the temperature range of (547 °C) to (695.9 °C), with peaks at (54.8 °C) and (695.8 °C) respectively. It was observed that in these last degradations, the losses are lower and are in the order of 0.18% and 0.27%, respectively, which indicates that they coincide with the final degradation of the trisodium citrate-capped gold nanoparticles. The total mass loss was equivalent to 18.9%. Similar results were obtained in previous studies [35].

According to the DSC curve (Figure 7), it was observed that the citrate-capped AuNP in the first heating showed endothermic peaks at (159.4°C), (76.4°C), (19.1°C), and (298.2 °C) and an exothermic peak at (323.7 °C) where the latter corresponds to the crystallization of trisodium citrate. The first endothermic peak corresponds to the evaporation of adsorbed water. The second and third peaks could correspond to the beginning of the degradation of NaCit. The endothermic peak at (298.2 °C) is related to the melting point of the NaCit that coats the surface of the gold nanoparticles. Similar results were obtained in previous studies by [36]. The XRD patterns of the AuNPs (Figure 8a) showed diffraction peaks at 2θ = 38.1°, 44.3°, 64.5°, and 77.5°. These values match the corresponding (111), (200), (220), and (311) lattice planes of AuNP and were in accordance with the Joint Committee on Powder Diffraction Standards (JCPDS) database no. 04-0784, confirming the formation of AuNP. According to the results obtained, it is clear that the AuNPs formed were crystalline in nature. Similar results were reported by [37] and [38]. The strong and high diffraction peak found here indicates the high crystallinity of the synthesized gold nanoparticles. To determine the average size of the sample crystals, the Debye–Scherrer formula was applied for the four diffraction peaks: (111) at 38.1°, (200) at 44.3°, (220) at 64.5°, and (311) at 77.5°, obtaining an average size of 15.4 nm. The diffraction patterns obtained demonstrated that the synthesized AuNPs were composed of pure crystalline gold nanoparticles. These results were in perfect agreement with the particle size obtained from the TEM analysis. Related results were reported by [39]. Furthermore, the appearance of several absorption peaks at 2θ = 9.1°, 11.2°, 17.5°, 18.6°, 19.9°, 23.9°, 25.1°, 27.3°, 31.6°, 33.4°, 35.4°, 45.3°, 56.5°, 66.1°, and 75.2°. All these absorption peaks correspond to the NaCit that covers the AuNP surface. Also, it was observed that the diffraction peaks localized at 2θ = 56°, 66°, and 75.2°, belonging to NaCit, showed a higher intensity in AuNP (Figure 8a) than in citrate alone (Figure 8b). This may indicate a weak influence on the AuNP surface to maintain the stability. 

As can be observed, the diffraction patterns (XRD) of NaCit used as a reducer and a stabilizer in the synthesis of gold nanoparticles are presented in Figure 8b, where it is observed that it presented several crystalline peaks. The most pronounced peaks were visualized at 2θ = 11.2°, 17.5°, 23.9°, 27.3°, 32.6°, 34.1°, 36.7°, 40.1°, 45°, and 46.3°; it is noted that the main peaks obtained were found in the XRD patterns of ICDD-00-056-0123 and correspond to the crystallinity of NaCit. The corresponding results were achieved in previous studies by [35]. Other lower intensity crystalline peaks were observed at 2θ = 48.5° to 58.5°. Finally, the crystalline peaks of minimum intensity perceived after 2θ = 60° are meaningless.

### 3.2. The Development and Characterization of Alg. Bead-Shaped Indicator Film

Figure 9 shows the results of the development of the indicator film (Alg. bead shape) using the casting technique, observing the complete formation of the film based on the gold nanoparticles and their in vitro evaluation with ethylenediamine, and observing the color changes after the reaction with the test molecule.

In the FTIR spectra of the Alg. beads (Figure 10), a large absorption band was observed in the range of 3600 to 3000 1/cm, related to the stretching vibration of the OH group and the C-H vibration bands at 2935 1/cm. The bands observed at 1594 1/cm and 1415 1/cm were attributed to the asymmetric and symmetric stretching vibrations of the COO^−^ groups of Alg., respectively, and are specific for ionic bonding. The shoulder located at 1080 1/cm, which was related to C–C and C–O stretching, can also be attributed to crossover. The absorption band detected at (1027 ± 4 1/cm) shows a higher intensity in relation to the 1080 cm^−1^ band, suggesting a stronger O-H binding vibration or a stronger binding of Ca^2+^ to the guluronic acids of Alg. In contrast, the stretching vibration bands observed at approximately 939 1/cm and 889 1/cm were specific to the guluronic and mannuronic acids. Also, small displacements of the carboxyl groups were observed, which may be indicative of an ionic union between Ca^2+^ and the Alg. chains [40]. In addition, in the evaluation of the colorimetric indicator with ETD, a crossover was observed in the stretching of the O-H group in the range of 3600–3000 1/cm. This was due to the reaction of the ETD with the carboxyl group of the NaCit on the surface of the gold nanoparticles, causing the nanoparticles to clump together. This led to the appearance of new bands in the indicated range.

On the other hand, it is known that Alg. is a polymer with a strong hydrophilic character [41], so its contact with the ethylenediamine solution and the modifications provided by the gold nanoparticles in its structure could produce a crossover.

In Figure 11 and Table 2, the four or five main stages or successive losses can be seen in the thermograms. In the first stage, the weight loss occurred at the beginning of the heat treatment, in the range of (47.8 °C) to (168.8 °C) which was mainly attributed to the loss of free water absorbed by Alg. and the indicator [42]. The second stage of thermal degradation was sensed at the onset of the temperature ranging from about (172.1 °C) to (259.3 °C) associated with the thermal degradation of the polymer as well as the polymer capping around the nanoparticles. The third and fourth weight losses could be related to the conversion of the remaining polymer into carbon residues [43] in the case of Alg.; however, in the colorimetric indicator, the third and fourth losses could correspond to the degradation of the citrate and the residual compounds present in the sample. It was observed that the colorimetric indicator with a weight loss of 47% presented certain thermal stability in relation to Alg. and its evaluation with ETD. This is because the addition of AuNP could have affected the stability of the Alg. beads due to the increase in the negative charge density in the hydrogel matrix [44]. Moreover, it was observed that there was a greater weight loss (54.7%) in the control Alg. with respect to the colorimetric indicator and its reaction with ETD, respectively, because Alg. is a hydrophilic polyanion and is sensitive to pH [45]. On the other hand, a weight loss in the order of 55.7% was observed in the colorimetric indicator evaluated with ETD. This weight loss could be due to the interaction of the amino group (NH2) of the ETD with the carboxyl group (COO^−^) of the oxygen-hungry NaCit on the surface of the AuNPs, which results in destabilization of the citrate and aggregation of the nanoparticles that could, to a certain extent, decrease the temperature resistance of the colorimetric indicator until it degrades above (700 °C). No significant differences were observed between the weights lost (*p* ≥ 0.05).

The DSC analysis curves reflect the thermal properties of the control Alg., the colorimetric indicator, and the colorimetric indicator evaluation with ETD. The endothermic melting peak of water crystallization appeared at 100.7, 103.6, and 106.9 °C in Alg., the colorimetric indicator, and the colorimetric indicator with ETD, respectively (Figure 12). The results obtained from the current experiment did not present significant differences (*p* ≥ 0.05) in any of the analyzed indicators nor in the control, with the exception of a greater broadening of the water absorption peak in the evaluated indicator with ETD, which could be due to the hydrophilic behavior of Alg. [46] and/or the formation of new chemical bonds and the interaction between the components of the in vitro assessed colorimetric indicator. Additionally, the enthalpy of fusion (ΔH) of the control (Alg.) (−536.7 J/g) was higher than that of the colorimetric indicator (−1217.8 J/g) and the colorimetric indicator with ETD (−556.9 J/g). This could indicate that AuNP with the analyte caused a reduction in this parameter. Chen et al., showed analogous results when evaluating the thermal property (DSC) of Alg. with the addition of thymol [47].

The XRD diffractograms of Alg., the colorimetric indicator, and the colorimetric indicator assessment with ETD are presented in Figure 13, where it was observed that Alg. (Figure 13a) presented diffraction peaks of lower intensity at 2θ = 31.5°, 45.2°, 56.4°, 66.2°, and 75.2°, corresponding to its amorphous nature. Similar results were reported in previous works by [48] and [49]. On the other hand, the colorimetric indicator (Figure 13b) showed two peaks of different intensities at 2θ = 31.7°, corresponding to Alg., and 2θ = 45.5°, which could be related to NaCit and/or Alg. because this same peak was observed with minimal variation in both. Furthermore, the peaks observed at 2θ = 22.4° and 38.3° belonged to NaCit and AuNP, respectively. Additionally, small peaks were observed at 2θ = 56.5°, 66.3°, and 75.2° associated with Alg. In other ways, the results obtained from the colorimetric indicator with ETD (Figure 13c) presented two peaks at 2θ = 31.7° and 45.3°. The first was associated with Alg. and the second may be associated with Alg. and/or NaCit. Likewise, two small peaks were visualized at 2θ = 22.5° and 29.4° related to NaCit. The other small peaks observed at 2θ = 56.4° and 75.2° were related to Alg. Additionally, two characteristic peaks of AuNP located at 2θ = 38.6° and 43.2° were observed, indicating the crystallinity of AuNP.

### 3.3. Determination of Total Volatile Basic Nitrogen (TVB-N)

Volatile compounds, such as trimethylamine, ammonia, and dimethylamine, produced by the destructive activities of microorganisms, are considered one of the most important parameters to determine the quality and freshness of fish and are known as the total volatile basic nitrogen (TVB-N) [50].According to the results (Figure 14), the initial values of TVB-N were 13.3 and 13.8 for the colorimetric indicator and the control, respectively, and they coincided correctly with the low initial microbial counts. These values are similar to the results obtained in previous research [51]; however, the TVB-N content increased progressively from 13.3 to 45.9 mg/100 g and from 13.8 to 44.8 mg N/100 g in the colorimetric indicator and the control, respectively, after 9 days of storage at 5 °C. This increase may be due to: (1) the activity of the spoilage bacteria that grow in the fish; (2) the enzymatic reaction that can occur in stored muscle food [52]; (3) and autolysis. A comparison of TVB-N values with colorimetric analysis showed that the color change of the colorimetric indicator was proportional to the TVB-N content and the pH change because, immediately after the death of the fish, a series of physical and chemical changes begin to occur in its body that led to its final alteration. Among these changes are mucus production on the body surface, rigor mortis, autolysis, organoleptic changes, and bacterial decompositions. The latter leads to the gradual increase in pH along with the TVB-N value and bacterial growth. All these parameters vary almost at the same time as the stored fish spoilage progresses; therefore, the increase in the amount of TVB-N caused an increase in pH and a color change in the colorimetric indicator.

### 3.4. Mesophilic Aerobic Count

The changes in aerobic bacteria counting in the salmon samples stored at 5 °C are shown in Figure 15. The initial Aerobic Mesophyll Count (AMC) values for the marine fish prior to cold storage typically range from 2 to 4 log_10_ CFU/g, and a value of 6 log_10_ CFU/g is considered the upper limit of acceptability [53]. Therefore, the values obtained in this study are within this range: 2.82 and 2.80 log_10_ CFU/g for the control and the sample with the colorimetric indicator, respectively. This indicates the freshness of the salmon with a low microorganism count. However, the bacteria grew gradually until reaching a value of 6.8 log_10_ CFU/g on the seventh day and reached values greater than 9 log_10_ CFU/g on the ninth day of storage at a temperature of (5 °C). This growth of microorganisms led to a greater production of biogenic amines, changes in the pH value, and the deterioration of the fish. However, there was no significant difference (*p* ≥ 0.05) in the growth of the bacteria between the control (fish sample only) and the sample with a colorimetric indicator inserted so that they grew at the same rate. Similar results were obtained when evaluating the spoilage potential of *P. fluorescens* in salmon at different temperatures [54]. According to previous studies, the bacteria that grow more at refrigeration temperatures are the so-called psychotropics. Also, Pseudomonas spp. was suggested as the main specific spoilage organism [55] and it is useful to predict shelf life with a cut-off level of 6.5 log_10_ CFU/g. Cheng and Sun reported that the main group of bacteria causing the spoilage of refrigerated or modified atmosphere vacuum-packed fish products are lactic acid bacteria, such as *lactobacillus*, *Streptococcus*, *Leuconostoc*, and *Pediococcus* spp. These bacteria are capable of spoiling foods by fermenting sugars and commonly cause undesirable defects, such as off-flavors, discoloration, gas production, slime production, and the lowering of pH [56]. In addition, they are capable of inhibiting the growth of other bacteria due to the formation of lactic acid and bacteriocins, and this facilitates their selective growth during fish spoilage [57].

### 3.5. Determination of pH

The changes in the pH values during storage are presented in Figure 16, where a decrease in the pH value was observed on day 2 of storage. This may be related to the production of some acidic substances because both the skin and the digestive system of fish can host a variety of bacteria, among which are lactic acid bacteria, which are facultatively anaerobic and grow very well under microaerophilic conditions [58]. These bacteria produce lactic acid as the main metabolic end product of carbohydrate consumption. When the lactic acid population increased during storage, there was an increase in the lactic acid capable of neutralizing the alkaline amine products, thus reducing the pH [57]. Undoubtedly, lactic acid bacteria predominated in the total natural microflora of the vacuum-packed fish fillets. On the other hand, another factor in the drop in pH could be the by-products of lipid oxidation, caused by the reaction of the amine compounds with the aldehydes. Similar results were obtained in previous investigations [53]; however, on days 7 and 9 of storage, variations in pH values from 7.54 and 7.92 to 9.45 and 9.60 were observed for the colorimetric indicator and the control, respectively. This was due to an increase in the growth of microorganisms in the muscle food, producing an increase in the TVB-N values and pH changes.

### 3.6. Indicator Color Measurement

The color parameters for the colorimetric indicators based on metallic nanoparticles were determined. According to Figure 17 and Table 3, *a** values decreased from 33 (redder) to 13 (greener) after 9 days of storage at refrigeration temperature. This means that the red color had a lower intensity at the end of the storage period. The values recorded for b* decreased from 13 to −24.11, indicating that the blue color appeared more pronounced at the end of the storage period. However, it was observed that the blue color began to appear from days 5 to 9 of storage; therefore, it is obvious that the blue color predominated in the evaluation of the colorimetric indicator. Researchers have reported that a ΔE* value greater than 4 can be easily detected with the naked eye, while values greater than 12 imply a complete color difference that is detectable even by untrained panelists [59]. Therefore, the ΔE* value of the colorimetric indicators obtained in this experiment could be detected by the human eye during fish storage. Generally, the indicators used as a colorimetric sensor exhibit a wide range of color variations depending on the pH, which can be affected by the TVB-N content during the storage period. This has the obvious advantage that color changes can be inspected with the naked eye to detect the rate of spoilage of the packaged fish. On the other hand, a progressive increase in the ΔE parameter was observed until the end of the storage time. This can be attributed to the loss of water that occurs as time passes, which gives rise to greater water deposits on the surface of the fish and inevitably leads to a variation in the value of this parameter.

When comparing the total color differences (ΔE) (Table 3) from the second to the last day of storage, it was observed that there were significant differences in the color of the colorimetric indicators analyzed using *p* ≤ 0.05.

### 3.7. Antibacterial Activity of AuNP

The antibacterial properties of AuNP can affect the effectiveness of the colorimetric indicator in the early stages of fish spoilage. However, according to the results obtained from the microbiological analyses (Mesophilic aerobic count), it can be observed that the bacteria grew progressively until the last day of storage, without presenting significant inhibition at the beginning. Nor was any reduction detected in the TVB-N values in the first days of storage, that means the amount of volatile compounds continued to vary on the second day of storage, going from 13.86 ± 0.5 to 16.87 and 16.67 mg of N/100 g on the indicator and the control, respectively. Furthermore, it was noticed that the growth of certain bacteria on the second day of storage reduced the pH values. Hence, it can be concluded that no antibacterial activity of gold nanoparticles was observed at this stage of development of the colorimetric indicator. This may be due to the particles’ size because the size and dispersion capacity are extremely necessary for good antibacterial activity of the nanoparticles. According to previous studies, it was confirmed that the size of the nanoparticles has great importance in modifying the cell membrane of microorganisms; therefore, it may affect their antimicrobial activity [60].

On the other hand, other researchers have indicated that citrate-capped gold nanoparticles did not have antibacterial activity when compared with kanamycin [61]. Also, AuNPs have negligible bactericidal effects at high concentrations [62]; therefore, in this study, the AuNP concentration used in the indicator was 0.3% *v*/*v*. Based on the results achieved, the synthesized AuNPs did not affect the effectiveness of the indicator in the first days of fish storage.

## 4. Conclusions

In this study, a new colorimetric indicator based on gold nanoparticles inserted in a biodegradable, non-toxic, and eco-friendly polymer was developed to monitor fish spoilage products. The colorimetric indicator showed the ability to successfully detect total volatile basic nitrogen in the last days of fish storage through a color change; therefore, it can be used to detect TVB-N in spoiled fish.

The ETD (similar to TVB-N) was successfully used as a test molecule in the in vitro evaluation of the detection capacity of AuNP and the colorimetric indicator. This allows us to quickly observe the discovery power of AuNPs and their possible application in smart packaging. The results obtained from the XRD of AuNP synthesized with trisodium citrate dihydrate indicated its crystalline nature and face-centered cubic structure. On the other hand, the semi-crystalline nature of Alg. was not observed in the diffraction pattern.

Given the progressive changes observed in the values of TVB-N, pH, mesophilic aerobic count (RAM), and color, it can be certified that the deterioration of the Atlantic salmon (*salmo salar*) occurred during the last days of storage at refrigeration temperature. Furthermore, significant changes were observed in bacterial growth above the allowable acceptability limit of 6.5 log10 CFU/g, as reported by some researchers. On the other hand, the values obtained in color determination, mainly the total color difference (ΔE), gradually increased over the storage time until a color difference visible to the naked eye was obtained. These results indicated that the colorimetric indicator showed its effectiveness in detecting fish spoilage products through a color change. In conclusion, the gold nanoparticles-based colorimetric indicator system is suitable for monitoring the quality of refrigerated fish at (5 °C).

## Figures and Tables

**Figure 1 polymers-16-00829-f001:**
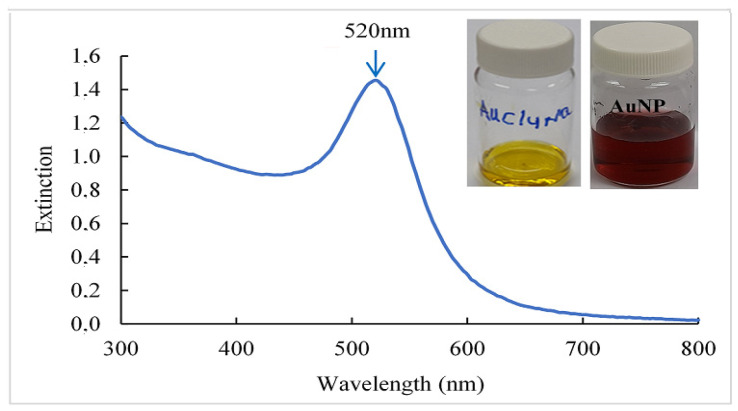
UV-VIS spectrum and photographs of gold nanoparticles (AuNP) obtained. Sodium tetrachloroaurate (NaAuCl_4.2_H_2_O) was used as a precursor of AuNP and NaCit (C_6_H_5_Na_3_O_7_·2H_2_O), as a reducer and a stabilizer.

**Figure 2 polymers-16-00829-f002:**
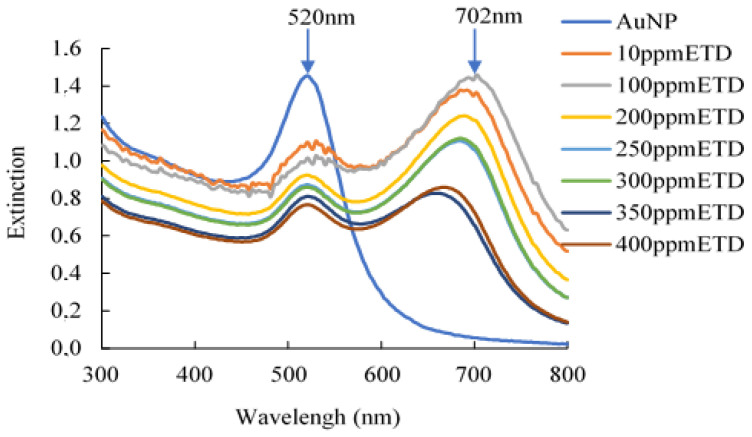
UV-VIS evaluation of the synthesized gold nanoparticles, based on different ETD concentrations (10–400 ppm).

**Figure 3 polymers-16-00829-f003:**
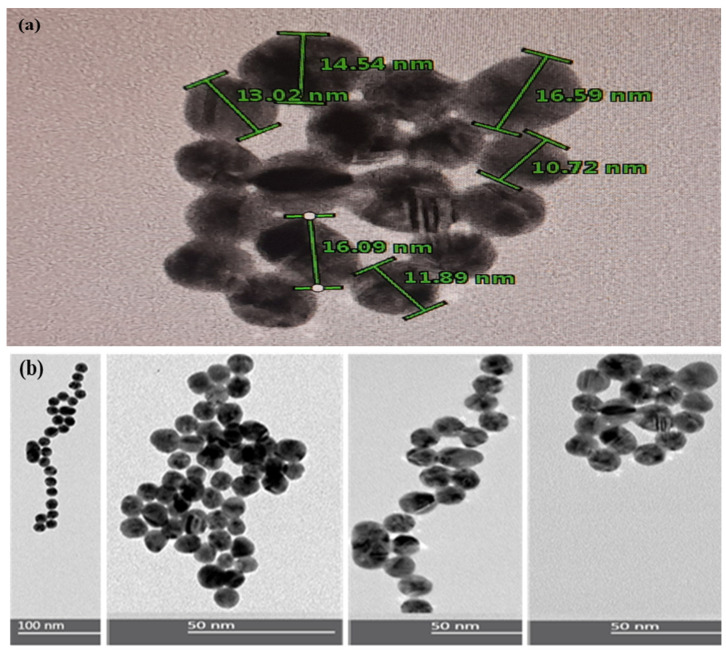
Images of (**a**) AuNPs and (**b**) AuNP+ ETD (aggregated form).

**Figure 4 polymers-16-00829-f004:**
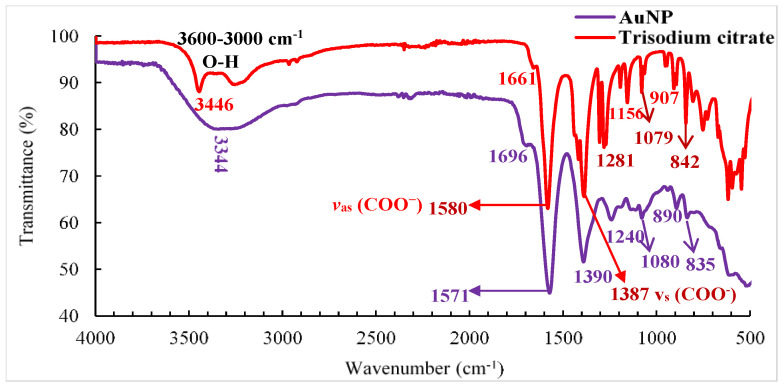
Infrared spectroscopy (FTIR) of AuNP functionalized with trisodium citrate.

**Figure 5 polymers-16-00829-f005:**
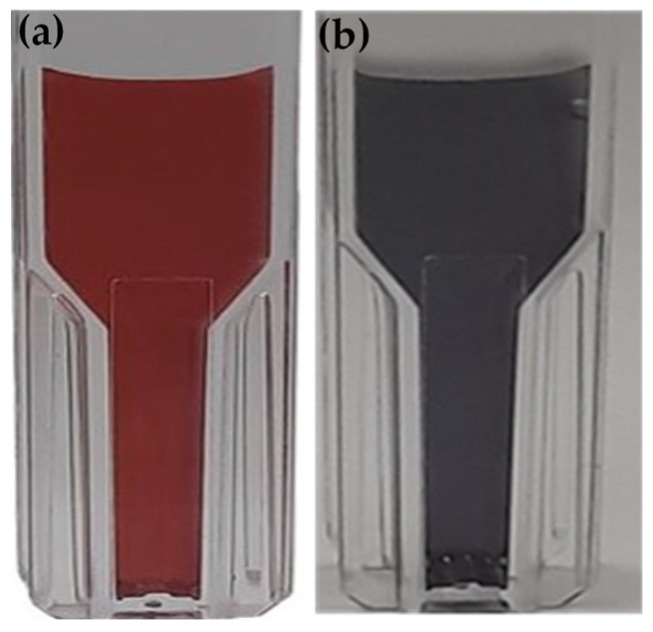
Colloidal solution of nanoparticles (**a**) AuNP (red) and (**b**) AuNP + ETD (black).

**Figure 6 polymers-16-00829-f006:**
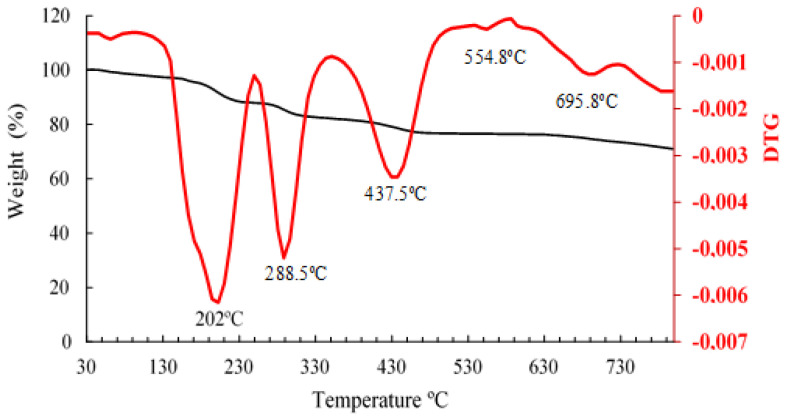
Thermogravimetric analysis TGA (black) and DTG (red) curves of citrate-capped AuNP.

**Figure 7 polymers-16-00829-f007:**
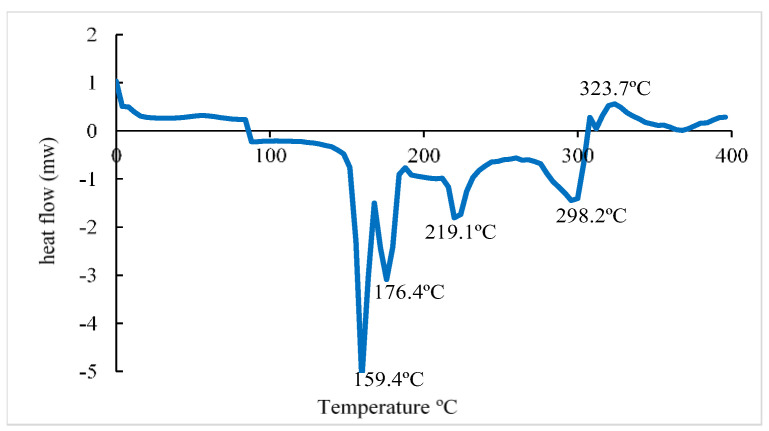
Differential scanning calorimetry analysis of gold nanoparticles (AuNP) capped by trisodium citrate.

**Figure 8 polymers-16-00829-f008:**
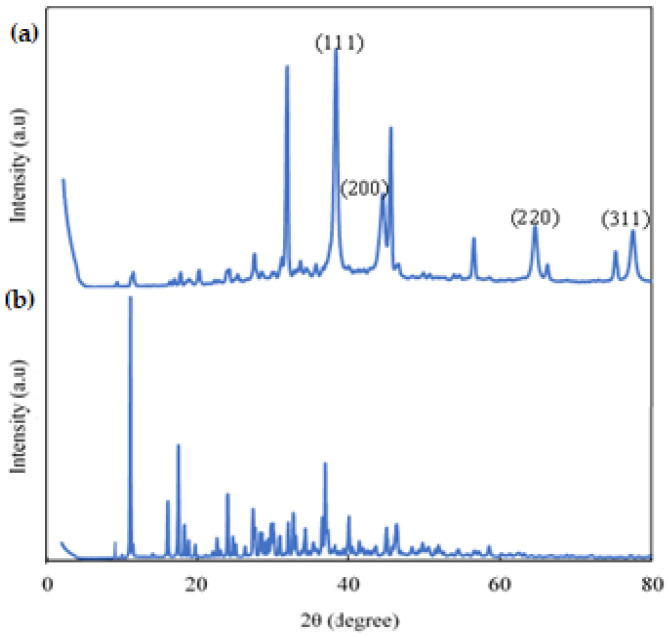
X-ray diffraction of (**a**) AuNP and (**b**) NaCit.

**Figure 9 polymers-16-00829-f009:**
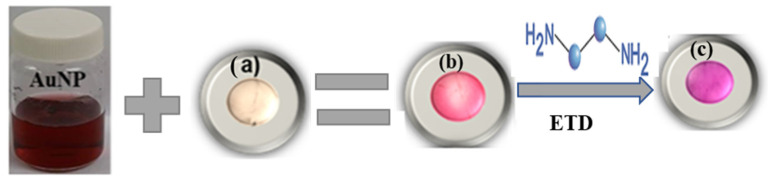
(**a**) Sodium Alginate (Alg.), (**b**) the colorimetric indicator (Alg/AuNP), and (**c**) the colorimetric indicator + ETD. AuNP: gold nanoparticles. ETD: Ethylenediamine.

**Figure 10 polymers-16-00829-f010:**
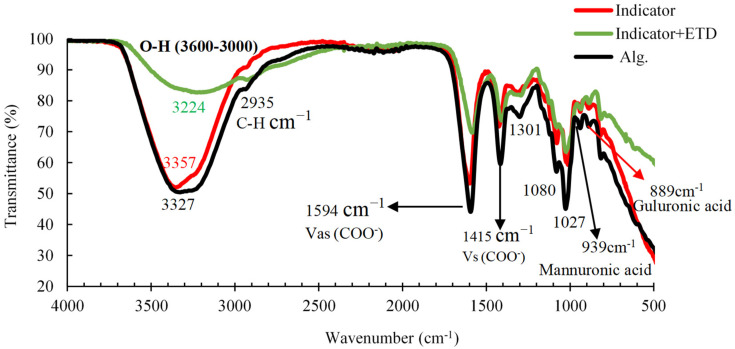
Infrared spectroscopy (FTIR) of the colorimetric indicator (Alg./AuNP), the colorimetric indicator + ETD, and Alg.

**Figure 11 polymers-16-00829-f011:**
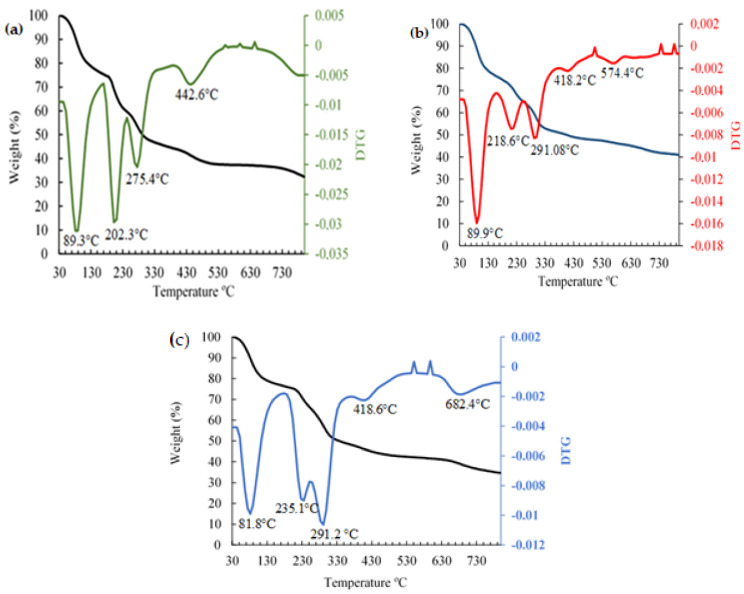
Thermogravimetric analysis of: (**a**) Alg. TGA–black DTG–green, (**b**) the colorimetric indicator (Alg./AuNP) TGA–black, DTG–red, and (**c**) the colorimetric indicator + ETD TGA–black, DTG–blue.

**Figure 12 polymers-16-00829-f012:**
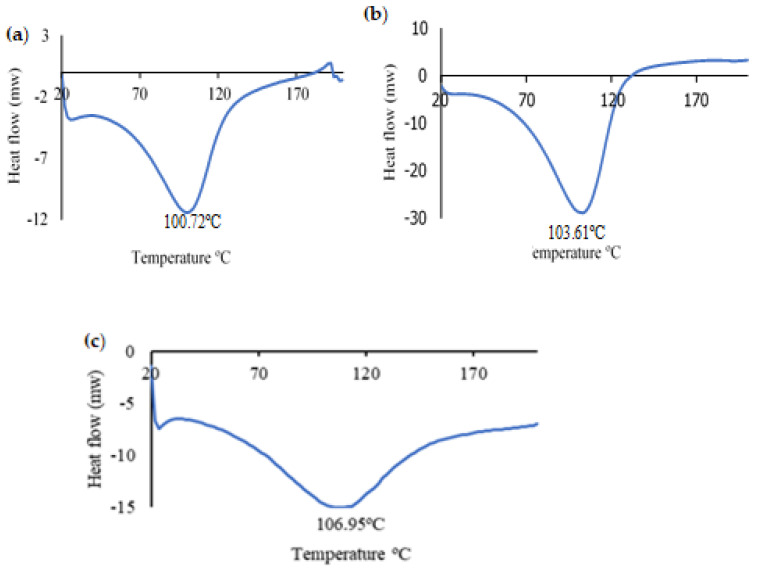
Differential scanning calorimetry DSC. (**a**) Alg., (**b**) the colorimetric indicator (Alg./AuNP), and (**c**) the colorimetric indicator + ETD.

**Figure 13 polymers-16-00829-f013:**
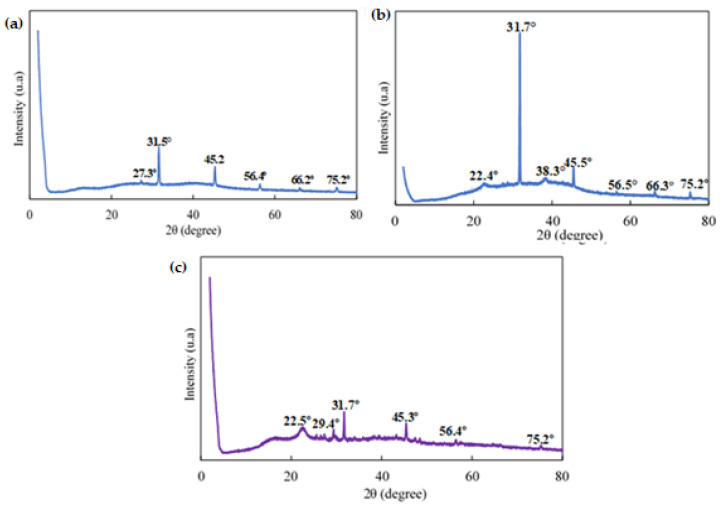
X-ray diffraction of: (**a**) Alg., (**b**) the colorimetric indicator, and (**c**) the colorimetric indicator + ETD.

**Figure 14 polymers-16-00829-f014:**
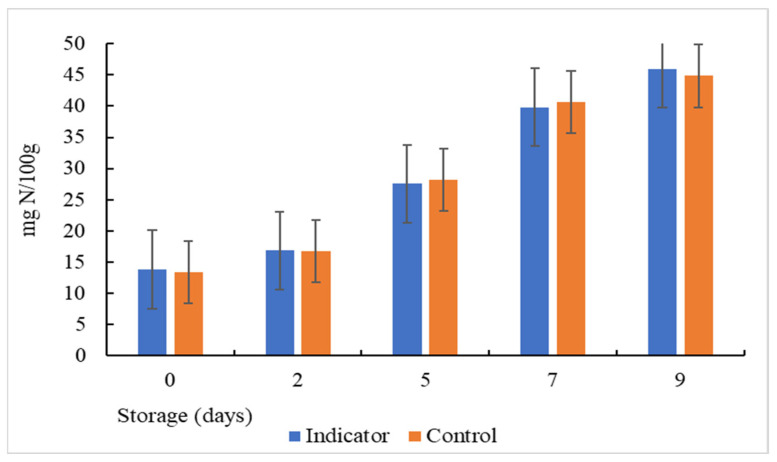
Determination of total volatile basic nitrogen (TVB-N) in fish stored at (5 °C).

**Figure 15 polymers-16-00829-f015:**
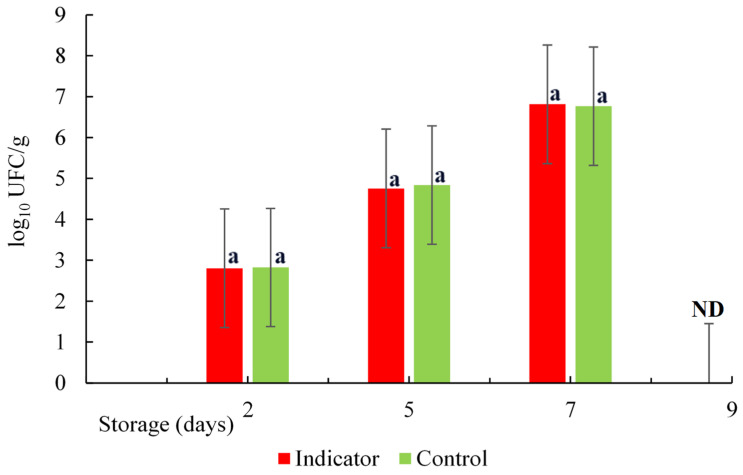
Count of mesophilic aerobes as a function of time. Same lower-case letters in each day indicate no statistically significant differences. ND: not determined.

**Figure 16 polymers-16-00829-f016:**
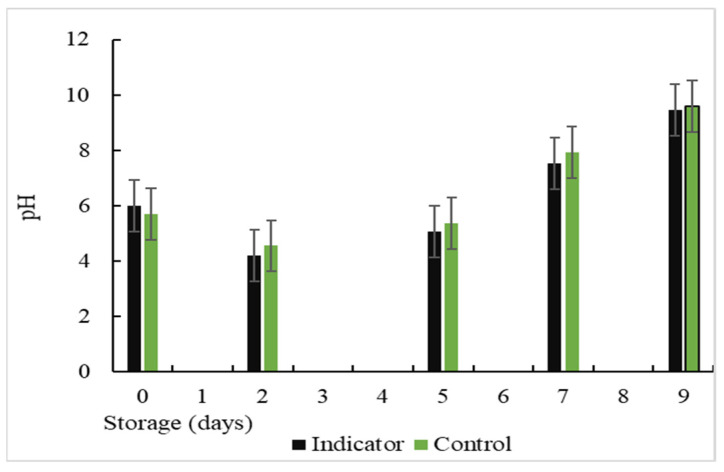
Changes in the pH value during storage at a temperature of (5 °C).

**Figure 17 polymers-16-00829-f017:**
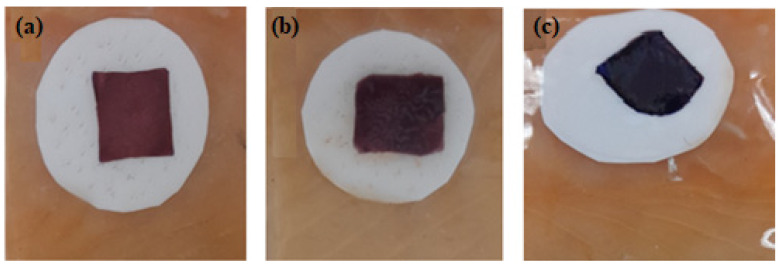
Color changes of the colorimetric indicator in fish stored at (5 °C). (**a**) Day 0, (**b**) day 7, (**c**) day 9.

**Table 1 polymers-16-00829-t001:** Color measurement of gold nanoparticles.

Color Parameters	AuNP	AuNP + ETD
*L**	10.56 ± 0.48	9.96 ± 0.40
*a**	24.47 ± 0.29	3.17 ± 0.24
*b**	13.32 ± 0.45	−6.06 ± 0.48
ΔE	–	28.81

*L** = lightness (0 = black, 100 = white), +*b** (yellow), −*b** (blue), +*a** (red), −*a** (green).

**Table 2 polymers-16-00829-t002:** TGA analysis results of Alg., the colorimetric indicator, and the colorimetric indicator + ETD.

Samples	Temperature (°C)	T-Onset (°C)	T50 (°C)	T Endset (°C)	Weight Lost (%)
Sodium Alginate (Alg.)	30–800	49.4	89.3	168.8	23.4 ± 1.3 ^a^
		172.8	202.3	243.7	15.8 ± 1.6 ^a^
		246.5	275.4	316.7	11.1 ± 0.4 ^a^
		388.3	442.6	460.3	4.4 ± 0.5^a^
Indicator (Alg. + AuNP)	30–800	47.8	89.9	153.9	22.2 ± 4 ^a^
		172.1	218.6	237.4	9.4 ± 0,7 ^a^
		239.1	291.08	314.4	12 ± 1.6 ^a^
		347.8	418.2	397.2	1.7 ± 0.3 ^a^
		527.8	574.4	620.8	1.7 ± 0.08 ^a^
Indicator + ETD	30–800	49.5	81.8	141.2	19.9 ± 4 ^a^
		207.1	235.1	259.3	19 ± 4.6 ^a^
		267.3	291.2	363.1	11.3 ± 0.4 ^a^
		375.2	418.6	445.5	3.6 ± 0.6 ^a^
		637.6	682.4	729.8	1.9 ± 0.3 ^a^

^a^: Same lower-case letters in the same row indicate no statistically significant differences.

**Table 3 polymers-16-00829-t003:** Indicator colorimetric analysis.

Storage Times (Days)	*L**	*a**	*b**	ΔE
0	17.86 ± 3.57	33.47 ± 4.09	13.01±3.94	0
2	9.60 ± 0.32	12.42 ± 2.09	0.8 0± 0.75	25.7
5	24.57 ± 0.87	12.18 ± 1.76	−8.05 ± 0.53	30.6
7	10.26 ± 0.12	13.39 ± 2.32	−19.11 ± 0.72	38.7
9	8.86 ± 0.61	13.73 ± 1.04	−24.11 ± 1.01	43.1

*L**: lightness (0 = black, 100 = white), *+b** (yellow), *−b** (blue), *+a** (red), −*a** (green). ΔE: total color difference.

## Data Availability

Data are contained within the article.

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
