# Peer review of "Colorimetric Indicator Based on Gold Nanoparticles and Sodium Alginate for Monitoring Fish Spoilage"

_polymers, 2024, doi:10.3390/polym16060829_

Round 1

Reviewer 1 Report

Comments and Suggestions for Authors

It is an interesting research focused on the utilization of gold nanoparticles and sodium alginate as colorimetric indicator for fish spoilage monitoring. The detection capacity of gold nanoparticles was evaluated using ethylenediamine (ETD) at different concentration and observed by UV-VIS spectroscopy and TEM microscopy. The AuNP + sodium alginate indicators were further physico-chemically investigated by XRD, FTIR and DSC. The prepared indicators were tested on packed salmon samples maintained at 5 degrees Celsius proving their efficacy in spoilage detection. The manuscript is well documented with relevant references and the experimental setup is well organized. Unfortunately there are some aspects that require revisions according detailed comments below:

Comment 1) The scan rate of 6 degrees/minute used for the XRD patterns registration is to high and might affect the peaks development especially for the gold nanoparticles. The scan rate of 1 degree/minute is strongly recommended.

Comment 2) XRD pattern of AuNP capped with sodium citrate is presented in a poor manner. The most representative peaks are truncated. The AuNP peaks are marked while the sodium citrate peaks are still unassigned. All peaks in the XRD patterns must be properly assigned.

Comment 3) It is recommended to apply Scherrer formula for the AgNP peaks to establish the average crystallite size of the gold nanoparticles and compared with TEM images.

Comment 4) AuNP observed in TEM images, Figure 3a, seems to have a heterogeneous internal structure. It might be an image artefact induced by low quality of the presented images. I would recommend presenting a larger observation field corresponding to a scale bar of 100 nm. Thus the AuNP will be more numerous and with a sharper aspect.

Comment 5) Figure 3b clearly evidences the AuNP clusterization tendency in presence of ETD. Unfortunately each of presented TEM images shows only a single cluster. However, it will be useful to present a TEM image at lower magnification (scale bar of 200 nm) to reveal few adjacent clusters.

Comment 6) The scan speed of XRD patterns for indicator and indicator + ETD in Figure 13 is totally inappropriate. These XRD patterns are completely useless because the AuNP peaks are completely missing. They are missed due to highly speed of 6 degrees/minute. It is mandatory effectuating new XRD patterns for these samples with a speed of 1 degree/minute to catch the AuNP peaks beside sodium alginate and ETD. All peaks must be propely assigned to their corresponding phase.

Comment 7) It will be useful presenting a TEM or SEM image of the indicator along with a discussion of its microstructure.

Comment 8) AuNP are known for their antibacterial properties. Their presence in the indicator might locally inhibit the bacterial proliferation in its close proximity. Does the Au NP antibacterial effect affects the indicator efficacy in the early stages of the fish spoilage? Fact should be discussed in a distinct paragraph.

Comment 9) Reference 41 is incomplete; it must be completed with all data required by the template.

Author Response

Dear Author 1, I think I respond correctly to the comments made in my paper. Except that I did not have the possibility to present a TEM or SEM of the colorimetric indicator analyzed. I hope that the answers provided in the paper respond to your questions and/or comments.
Thank you so much.

Reviewer 2 Report

Comments and Suggestions for Authors

The authors present a method for “by-eye” monitoring fish spoilage with the use of Au NPs introduced into a polymer. The topic is relevant and interesting, and lies within the journal scope.

However, the paper is too long and difficult for readers. Some data are surplus: X-ray diffraction, DSC, TGA/DTGA do not provide any information regarding the colorimetric approach being studied.

The main point that needs clarification: Figure 17 presents photos for days 0, 7 and 9, while Table 3 contains the colorimetric analysis for days 0, 2, 5, 7, and 9. And from this table it is seen that the deltaE changes only slightly for days 2, 5 and 7, while the changes in aerobic bacteria count, pH, and TVB-N content (Figures 14-16) between days 2 and 7 are essential.  Is it possible to visually distinguish the difference in colors on days 2 and 7? Otherwise, this method cannot be used for “by-eye” monitoring.

It would be useful to present separate dependences of the total color difference on TVB-N content, pH and the number of aerobic bacteria. When three factors act together it introduces uncertainty into the results.

In Table 1 the total color difference for NPs with ETD added is presented, but from this table it is not clear, what the concentration of ETD is used in the experiment, and what the total color difference would be for a different concentration.

For me it remained unclear what connection does the ETD have with the fish spoilage?

Minor comments:

1.       Figure 9, what do the top and bottom parts of the image correspond to? They look the same.

2.       Paragraph 3.3. Transmission electron microscopy (TEM) appears twice.

3.       The paper has too many sub-paragraphs and it could be shortened.

Author Response

Dear Author 2, I think I respond correctly to the comments made in my paper.

Thank you so much.

Please cancel the first response sent because it was not complete. The second document has the complete answers. Thank you

Round 2

Reviewer 1 Report

Comments and Suggestions for Authors

All requested: corrections, completions and explanations were properly added to the manuscript.

Author Response

Before accepting your manuscript, I ask you to respond to the following comments and 
suggestions:
1. The initial suggestion from Reviewer 1, was to redo the x-rays at a slower scan rate (1° per 
minute). Did the authors really repeat the x-ray experiments, or did they just change the scan 
rate value in the experimental part (lines 150-151)? I ask this, because the diffractograms of 
figure 8 in the two versions of the manuscript are the same.
A. The XRD of AuNP in figure 8 was done at a scanning speed (1º/minute) as you already recommended. The results obtained with this scanning speed are almost the same as the  previous ones except for a slight difference in the intensity of the peak located at 2 theta  equal to 38.1º corresponding to position (111). I think that it is slightly different from the  first XRD of AuNP presented.

2. Figures 8 and 8a should be renamed as figure 8a and 8b, respectively. Second, these two figures must be presented on the same scale for comparison purposes, since the authors refer  to the intensity of the diffraction peaks between both figures. When I refer to equality of scales, I  mean that they can be plotted on the same graph, and not to change the size of the figure. The 
way the indexing of the diffraction peaks is done is not conventional. Please review the literature  to do proper indexing, and correct in both figures
A. Yes I did it.
3. In the revised version of the manuscript, please check the authenticity of figure 3, specifically  the middle image. I mean that in this image three types of gold particles are observed, which  are the same as each other. This appears to be a modified or edited image, and not real Au particles. The information provided by the images in 3a and 3b is sufficient to demonstrate the 
size of the gold NPs.
A. I do as requested.
4. On line 499, the authors mention "Nacit", which I assume refers to sodium citrate. Please 
correct.
A .NaCit corrected.
5. In section 3.13 (lines 717 to 749) of the revised version of the manuscript, the authors make  some imprecise comments, so I suggest that this entire section be revised again. For example,  but do not limit to the following:
- The semicrystalline nature of sodium alginate is not observed in the diffraction pattern. Commonly amorphous or semi-crystalline materials have a broad peak with low intensity.
- The authors refer to alginate as "Alg." "alginate" or "sodium alginate". I recommend that a  single name be used throughout the document for each of the materials referenced.
- The comment on lines 740-741 "The latter caused aggregation of the nanoparticles, and  probably, a slight shift of the AuNP diffraction peaks as seen here.", it is imprecise. I do not  agree with the authors that the aggregation of the particles causes the disappearance of the  gold diffraction peaks, nor that the aggregation causes the displacement of one of these peaks.  Please review and correct.
- On lines 745-747, the authors mention "which indicates that the incorporation of the test  molecule shows a weak interaction in the microstructure of the alginate". This comment is also  imprecise, since the XRD technique is not used to determine "interactions" between molecules.  Please review and correct.
A. I review and make appropriate corrections.

6. The figure on the first page, I want to understand that it is the graphical abstract. In any case,  it is not common for any figure to be included in this section or part of the manuscript. Please  remove it or accommodate it in the corresponding section of the work. In the case that it is the graphical abstract, it is loaded individually on the MDPI Susy platform, in the section with that name.

A. I put it in the corresponding section. Thank you 

Reviewer 2 Report

Comments and Suggestions for Authors

Dear authors! Although the article still seems too long to me, I think it can be published in its current form.

Author Response

  1. Regarding the comments of Reviewer 2, I consider that these are also very accurate. The responses given by the authors to Reviewer 2 enrich the discussion of the article. So, I ask the authors to try to include part of these responses in the manuscript, since some of them were included only in the response letter to the Reviewer. 
    A. I Include part of the answers given to Author 2 in the AuNP characterization section on lines 245-248 and 868-871.
  2. I agree with Reviewer 2's last comment "The paper has too many sub-paragraphs and it could be shortened." Although the authors did delete some subsections of the manuscript, it is still lengthy and, even more so, messy. I mean that in the results two sections of each characterization technique are presented. For example, sections 3.1 to 3.8 deal with the characterization of Au NPs. I suggest that the authors merge all these sections into one, and summarize the results, presenting only the most important ones. Likewise for sections 3.9 to 3.13, which deal with the characterization of the alginate film with Au NPs, I suggest that they also be merged and summarized.   A. i  shorten subparagraphs to facilitate better understanding of the article.
  3. Finally, the conclusions of the revised version of the manuscript are very short and do not help to highlight the importance and contribution of their work. Please rewrite this section.  A- I rewrote this section as you asked.
